# Extremely Polysubstituted Magnetic Material Based on Magnetoplumbite with a Hexagonal Structure: Synthesis, Structure, Properties, Prospects

**DOI:** 10.3390/nano9040559

**Published:** 2019-04-06

**Authors:** Denis A. Vinnik, Vladimir E. Zhivulin, Evgeny A. Trofimov, Andrey Y. Starikov, Dmitry A. Zherebtsov, Olga V. Zaitseva, Svetlana A. Gudkova, Sergey V. Taskaev, Denis S. Klygach, Maxim G. Vakhitov, Elena E. Sander, Darya P. Sherstyuk, Alexey V. Trukhanov

**Affiliations:** 1Laboratory of Single Crystal Growth, South Ural State University (National Research University), 454080 Chelyabinsk, Russia; zhivulinve@mail.ru (V.E.Z.); tea7510@gmail.com (E.A.T); Starikov-andrey@mail.ru (A.Y.S.); zherebtsov_da@yahoo.com (D.A.Z.); zaitcevaov@susu.ru (O.V.Z.); svetlanagudkova@yandex.ru (S.A.G.); s.v.taskaev@gmail.com (S.V.T.); vasia-ivanov-88@mail.ru (E.E.S.); daryasherstyuk77@gmail.com (D.P.S.); truhanov86@mail.ru (A.V.T.); 2Center of shared research facilities, Moscow Institute of Physics and Technology (State University), 141701 Dolgoprudny, Russia; 3Faculty of Physics, Chelyabinsk State University, 454001 Chelyabinsk, Russia; 4Department of Electronic Materials Technology, National University of Science and Technology “MISIS”, 119049 Moscow, Russia; 5Department of Design and Manufacture of Radio Equipment, South Ural State University (National Research University), 454080 Chelyabinsk, Russia; klygachds@susu.ru (D.S.K.); max_v_333@mail.ru (M.G.V.); 6Institute of Radioelectronics and Information Technologies, Ural Federal University Named after the First President of Russia B.N. Yeltsin, 19 Mira Street, 620002 Ekaterinburg, Russia; 7Laboratory of Magnetic Films, Scientific and Practical Materials Research Centre of NAS of Belarus, 220072 Minsk, Belarus

**Keywords:** high-entropy phase, magnetoplumbite structure, inorganic compounds, magnetic materials, crystal growth, crystal structure, magnetic properties

## Abstract

Crystalline high-entropy single-phase products with a magnetoplumbite structure with grains in the μm range were obtained using solid-state sintering. The synthesis temperature was up to 1400 °C. The morphology, chemical composition, crystal structure, magnetic, and electrodynamic properties were studied and compared with pure barium hexaferrite BaFe_12_O_19_ matrix. The polysubstituted high-entropy single-phase product contains five doping elements at a high concentration level. According to the EDX data, the new compound has a formula of Ba(Fe_6_Ga_1.25_In_1.17_Ti_1.21_Cr_1.22_Co_1.15_)O_19_. The calculated cell parameter values were *a* = 5.9253(5) Å, *c* = 23.5257(22) Å, and *V* = 715.32(9) Å^3^. The increase in the unit cell for the substituted sample was expected due to the different ionic radius of Ti/In/Ga/Cr/Co compared with Fe^3+^. The electrodynamic measurements were performed. The dielectric and magnetic permeabilities were stable in the frequency range from 2 to 12 GHz. In this frequency range, the dielectric and magnetic losses were −0.2/0.2. Due to these electrodynamic parameters, this material can be used in the design of microwave strip devices.

## 1. Introduction

Starting from 2010 [1], the properties of high-entropy oxide phases have been intensively studied [2,3,4,5,6,7,8,9,10]. Primarily, cubic oxide phases formed by divalent metals Mg, Co, Ni, Cu, and Zn [4,7,8], or rare earth metals were studied [9,10]. High-entropy oxide nanotubes were also reported [11], but this material is hard to consider as stable crystalline phase. Next, high-entropy phases with a more complex crystal structure-like spinel [12] or (Al,Cr,Ti)FeCoNiO_x_ [13] were achieved. Two papers describe high-entropy phases with the perovskite structure [14,15].

Some researchers have studied the electrophysical and magnetic properties of high-entropy oxide phases [3,5,6,13,16,17]. It was shown that Ti_x_FeCoNiO_y_ films have extremely low-electrical resistance values [17]. Some authors [3] declared an exceptionally high value of the dielectric constant of materials formed by Mg, Co, Ni, Cu, Zn, Li, and Ga oxides. The following studies on such materials led to the discovery of phases (Mg,Co,Ni,Cu,Zn)_1-x-y_Ga_y_A_x_O (with A=Li,Na,K) with high-ionic conductivity, which makes them promising for use as solid electrolytes [5] that could be obtained as a thin film [16]. A recently published paper aimed to generalize the theoretical description of high-entropy phase stability [18].

Still, there were no reports on high-entropy oxide materials with the magnetoplumbite structure PbFe_12_O_19_, which are widely used in modern technology thanks to a combination of its electrophysical and magnetic properties [19,20,21,22,23,24,25,26].

Traditionally, the barium and strontium hexaferrites have been explored as the parent phases in which iron atoms are replaced by a certain amount of one or two other elements. The introduction of dopants makes it possible to manage the electrical and magnetic properties of the resulting phase. This opens up the possibility of obtaining materials whose electrical and magnetic characteristics exactly correspond to the requirements of electronic devices [19]. Synthesis of high-entropy phases with the magnetoplumbite structure can be considered as the next step in obtaining substituted structures, which allow regulating the properties of materials in wider limits [27].

This work is devoted to studying the possibility of synthesizing a high-entropy phase with a magnetoplumbite structure, whose composition can be described as BaFe_6_(Ti,Co,In,Ga,Cr)_6_O_19_, studying its structure, and magnetic and electrophysical properties.

## 2. Materials and Methods

The set of elements used for the synthesis was selected on the basis of the following considerations. For the parent “BaB_12_O_19_” structure, the doping elements of “B” type include Ti and In (that both proved to have a high ability to substitute iron [28,29]), and the elements that have similar ionic radii—Ga, Co, and Cr (Table 1).

In the target composition, half of the atoms of the “B” type were iron atoms, and the second half consisted of doping elements in equimolar concentrations. The configurational entropy of mixing in this case was lower than for the case when the concentrations of all elements, including iron, were equal. Thus, for the Ba(Fe_2_Ti_2_Co_2_In_2_Ga_2_Cr_2_)O_19_ phase, the configurational entropy of mixing would be equal to 1.792 R, whereas for our composition, Ba(Fe_6_Ti_1.2_Co_1.2_In_1.2_Ga_1.2_Cr_1.2_)O_19_ was 1.498 R.

However, reducing the iron content below that of Ba(Fe_6_Ti_1.2_Co_1.2_In_1.2_Ga_1.2_Cr_1.2_)O_19_ led to the disappearance of useful electrophysical and magnetic properties of the material obtained. Therefore, the composition under study represents a compromise between the need to preserve the magnetic characteristics of magnetoplumbite and to obtain a sufficiently high-configurational entropy.

The solid-state sintering was used for crystalline high-entropy single-phase with a magnetoplumbite structure. The BaCO_3_, Fe_2_O_3_, TiO_2_, In_2_O_3_, Ga_2_O_3_, Cr_2_O_3_, and CoO (Russia, Yekaterinburg, Ural plant) were used as initial charge components. The mixture was ground in an agate mortar and filled into a 30-mL platinum crucible. Table 2 presents the initial weight ratios of charge compositions. The synthesis temperature that provides the single-phase formation was up to 1400 °C. The synthesis time was up to 5 h.

The sample morphology and chemical composition were investigated using a scanning electron microscope, Jeol JSM7001F (Tokyo, Japan), with an energy dispersive spectrometer, Oxford INCA X-max 80, for elemental analysis. Average composition and deviation were measured from 5–7 single-phase areas from each sample.

X-ray powder diffraction analyses were performed on a diffractometer Rigaku Ultima IV (Tokyo, Japan) in the 2Theta angular range from 10° to 80° with the speed of 2°/min. For this purpose, the samples were thoroughly powdered and then applied on a thin layer of single-crystalline silicon.

For the electrodynamic properties determination, the powder particles of the studied material had a size of no more than 0.1 mm. Due to this, the space between the outer and inner conductors of a filled coaxial line segment and the probability of void appearance, which may affect the measured parameters, decreased. The test material in the coaxial line was clamped on both sides, with polystyrene rings. More details about the measurement technique can be read elsewhere [31].

Since the values of polystyrene permittivity and permeability in the frequency range 2–12 GHz are ε ~ 1.1 and μ ~ 1 [32], their influence can be neglected in further calculations.

To account for the influence of polystyrene rings on the measured S-parameters, the following measurements were performed:-measuring S-parameters of the air-filled coaxial line with polystyrene rings;-measuring S-parameters of the material-filled coaxial line with polystyrene rings.

When filling the coaxial line segment with samples of the material, the weight of the filling substance was controlled. The weight of the studied samples was 7.3 ± 0.1 g.

The use of a powder material makes it possible not to make samples of a certain shape, but to fill the volume required for measurement with a fine-grained structure.

When measuring S-parameters of the coaxial line with the test material as a dielectric, a dual-port (Thru-Reflect-Line) TRL-calibration was applied. Calibration Kit RPC-N, 50 Ω, LRL Version manufactured by Rosenberger (Tittmoning, Germany), an additional airline (type N Beadless Air Line) 2553T15 manufactured by Maury (Ontario, California, USA), and measuring adapters were used as calibration standards. Measurement was performed using a single-port vector reflectometer CABAN 180-02 (Chelyabinsk, Russia).

To calculate the dielectric and magnetic permeability, the following measurements were made. In the first dimension, a segment of the coaxial line with the material was connected to the reflectometer, a load was set at the other end of the segment. In this case, measurements were made of the real and imaginary parts of the input resistance. Then, instead of the matched load, a short circuit was established, and the real and imaginary parts of the input resistance was measured.

## 3. Results

This section provides a concise and precise description of the experimental results of the morphology, chemical composition, crystal structure, magnetic, and electrodynamic properties investigation.

### 3.1. Morphology and Chemical Composition

The obtained crystalline high-entropy single-phase products with a magnetoplumbite structure were obtained within the μm-range grains. Typical SEM images of the free surfaces of the samples (Figure 1) illustrate the sizes and shapes of the crystals. Presumably, the hexagonal-shaped crystals can be associated with a magnetoplumbite phase. For all samples, the EDX analysis of 10 individual hexagonal crystals and sample areas was performed. The chemical composition and average sample formulas are given in Table 3. The special procedure was used for sample preparation for quantitative SEM/EDX analysis. To improve the measurement accuracy, the sample powders were applied to the holder and flattened with a glass. The standard deviations of the analyses were up to 5%.

### 3.2. Crystal Structure

From the PXRD data, it was concluded that the high-entropy sample was the single magnetoplumbite phase with the general formula BaFe_6_M_6_O_19_ (M—sum of dopant cations). The single-phase sample pattern is shown in Figure 2. The space group was established. The substituted sample and initial matrix have the same space group P6(3)/mmc. Calculated unit cell parameters of experimental pure matrix BaFe_12_O_19_, high-entropy sample Ba(Fe_6_Ga_1.25_In_1.17_Ti_1.21_Cr_1.22_Co_1.15_)O_19_, and literature data are presented in Table 3.

Unit cell determination from powder diffraction (Table 4) already indicated a substitution influence of Fe by Ti/In/Ga/Cr/Co. Due to the different ionic radius of Ti/In/Ga/Cr/Co compared with Fe^3+^ (r(Fe^3+^) = 0.55 Å; r(Ti^4+^) = 0.605; r(In^3+^) = 0.80 Å; r(Ga^3+^) = 0.62 Å; r(Cr^3+^) = 0.615 Å; r(Co^2+^) = 0.73 Å with CN = 6 [30]), and an increasing unit cell with dopants appearing was expected.

### 3.3. Electrodynamic Properies

The results of measuring the real and imaginary parts of the dielectric permittivity of the frequency of the samples of materials are present in Figure 3 and Figure 4. Sample 1 is initial matrix BaFe_12_O_19_ and Sample 2 is substituted sample Ba(Fe_6_Ga_1.25_In_1.17_Ti_1.21_Cr_1.22_Co_1.15_)O_19_ (numbers 1 and 2 at Table 2, Table 3 and Table 4).

The results of calculations of the dependence of the real and imaginary parts for magnetic permeability on the frequency of the material samples are present in Figure 5 and Figure 6.

The calculated dependences of the electric loss tangent and magnetic loss tangent on the frequency of the material samples are present in Figure 7 and Figure 8.

The calculated dependences of the skin layer on the frequency of the material samples are present in Figure 9.

It can be concluded that the electrodynamic parameters’ change in the polysubstituted material was not significant. In the frequency range from 10 to 18 GHz, the real part of the dielectric permittivity decreased for the Ba(Fe_6_Ga_1.25_In_1.17_Ti_1.21_Cr_1.22_Co_1.15_)O_19_ and increased for the BaFe_12_O_19_. The real and imaginary parts of the magnetic permeability have now significant changed and are very close for both samples. The dielectric loss tangents for these samples almost coincide, but the increase in losses at low and high frequencies should be noted for the substituted sample. The decrease in the thickness of the skin layer for the polysubstituted material was due to the increase in dielectric and magnetic losses (Figure 7, Figure 8 and Figure 9).

## 4. Conclusions

In the present work, a high-entropy single-phase product with a magnetoplumbite structure was obtained. The calculated brutto formula of substituted phase was Ba(Fe_6_Ga_1.25_In_1.17_Ti_1.21_Cr_1.22_Co_1.15_)O_19_. From the PXRD data it was concluded that the high-entropy sample was the single magnetoplumbite phase. The increased cell parameters values of the substituted sample were observed. The calculated cell parameter values were *a* = 5.9253(5) Å and *V* = 715.32(9) Å^3^. The electrodynamic investigation was performed. The dielectric and magnetic permeabilities were stable in the frequency range from 2 to 12 GHz. In this frequency range, the dielectric and magnetic losses were −0.6/1.2. Due to these electrodynamic parameters, this material can be used in the design of microwave strip devices.

## Figures and Tables

**Figure 1 nanomaterials-09-00559-f001:**
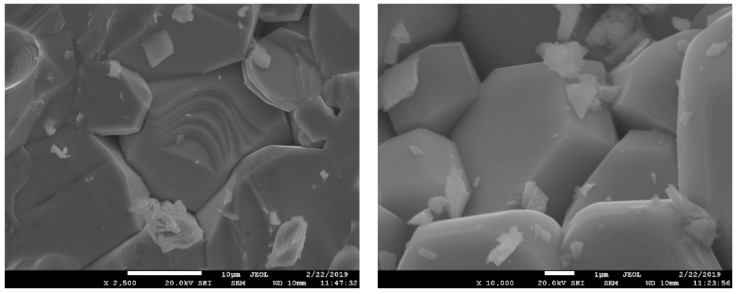
SEM images of a high-entropy sample with a magnetoplumbite structure.

**Figure 2 nanomaterials-09-00559-f002:**
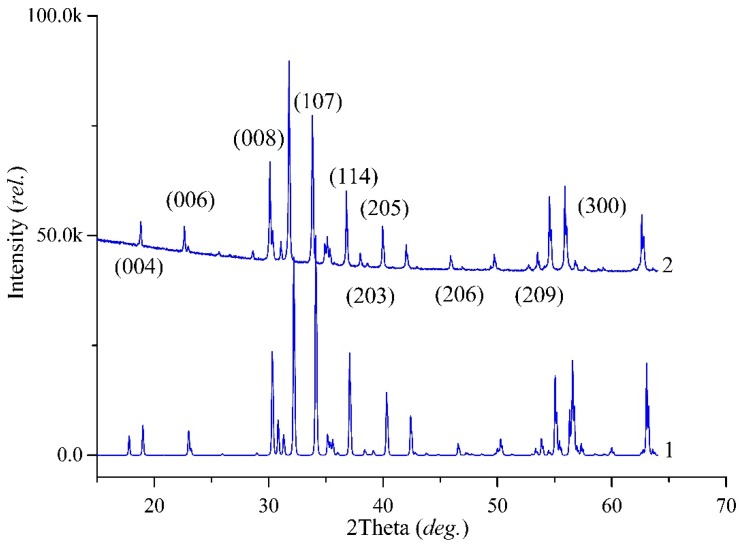
Powder XRD patterns: 1—simulated according to crystal structure data in the literature [33]; 2—experimental sample of Ba(Fe_6_Ga_1.25_In_1.17_Ti_1.21_Cr_1.22_Co_1.15_)O_19_. Differences arose due to the minor variations in intensities according to the high-substitution levels of Fe by Ti/In/Ga/Cr/Co and mostly different degrees in the preferred orientation of the grains.

**Figure 3 nanomaterials-09-00559-f003:**
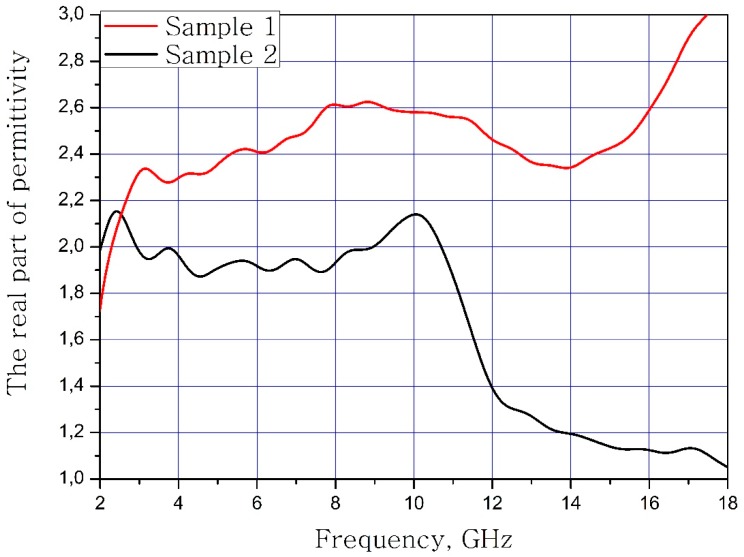
Frequency dependence of the real part of the dielectric permittivity of the samples.

**Figure 4 nanomaterials-09-00559-f004:**
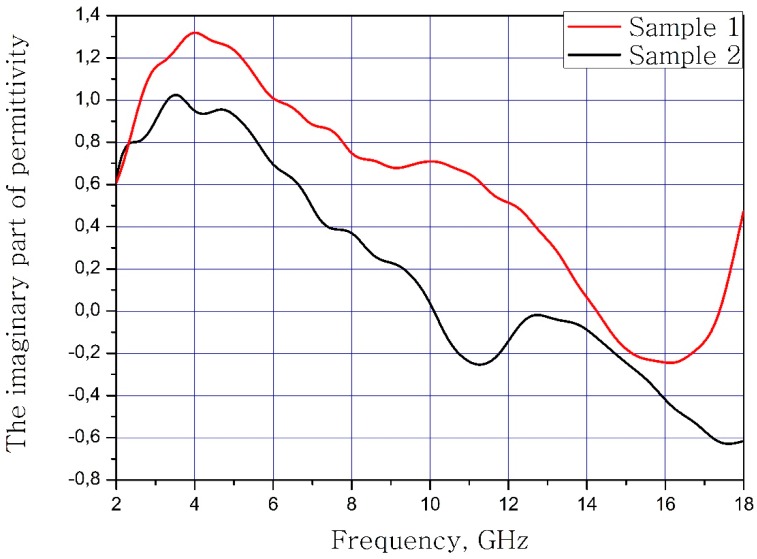
Frequency dependence of the imaginary part of the dielectric permittivity of the samples.

**Figure 5 nanomaterials-09-00559-f005:**
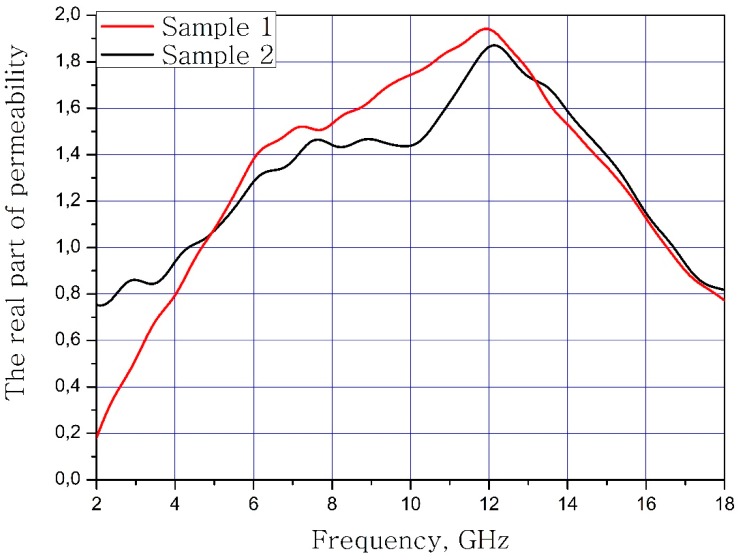
Frequency dependence of the real part of the magnetic permeability of the samples.

**Figure 6 nanomaterials-09-00559-f006:**
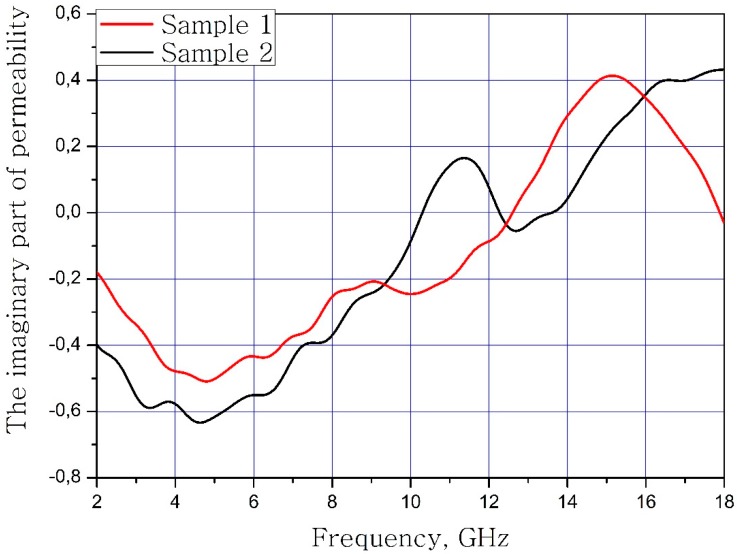
Frequency dependence of the imaginary part of magnetic permeability of samples.

**Figure 7 nanomaterials-09-00559-f007:**
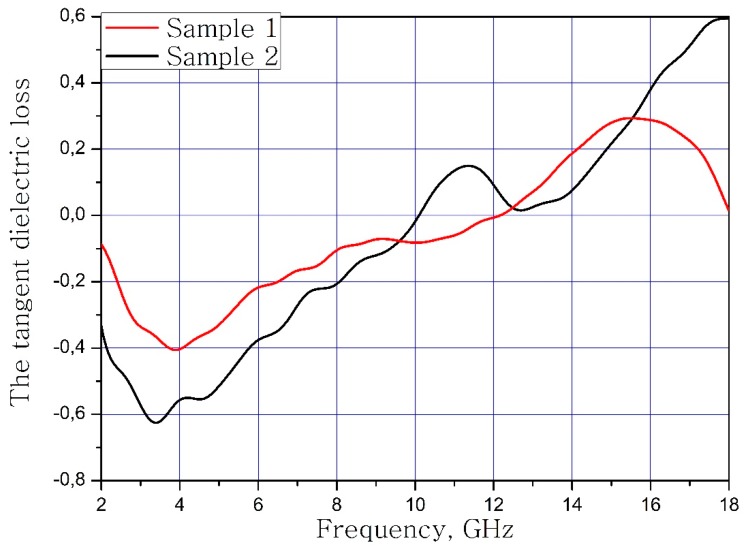
Frequency dependence of the dielectric loss tangent of samples.

**Figure 8 nanomaterials-09-00559-f008:**
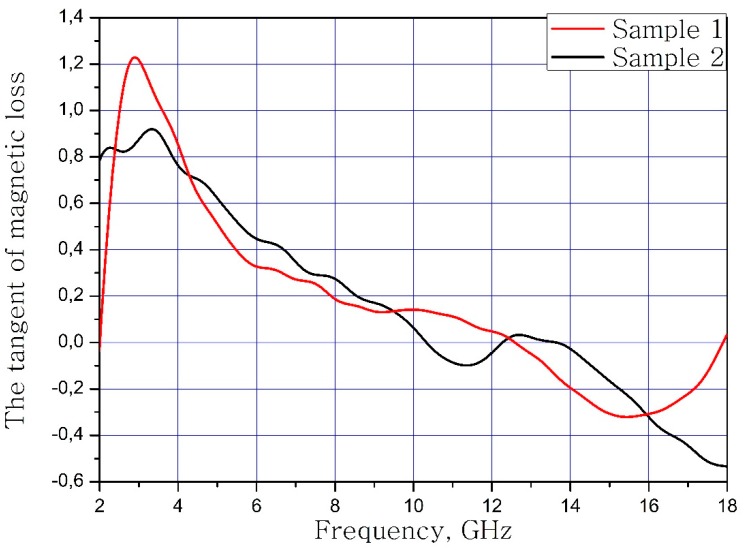
Frequency dependence of the tangent of magnetic loss samples.

**Figure 9 nanomaterials-09-00559-f009:**
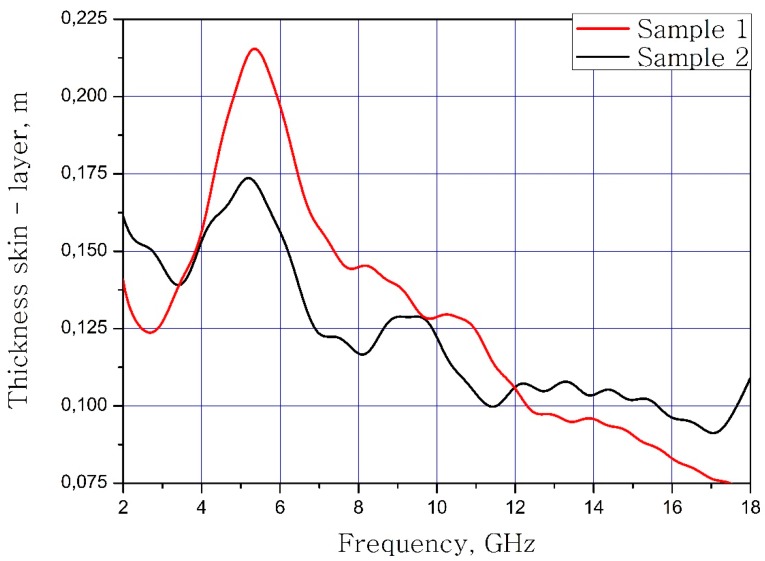
The frequency dependence of the thickness of the skin layer of samples.

**Table 1 nanomaterials-09-00559-t001:** The radii of B-type ions (CN6, low/high spin state) [30].

B	In^3+^	Ga^3+^	Cr^3+^	Fe^2+^	Ti^4+^	Fe^3+^	Co^3+^
ionic radii (pm)	79	62	61.5	61/77	60.5	55/64.5	52.5/61

**Table 2 nanomaterials-09-00559-t002:** Initial weight ratios of charge compositions (wt.%).

#	Target Composition	BaCO_3_	Fe_2_O_3_	TiO_2_	In_2_O_3_	Ga_2_O_3_	Cr_2_O_3_	CoO
1	BaFe_12_O_19_	17.078	82.922	-	-	-	-	-
2	Ba(Fe_6_Ga_1.2_In_1.2_Ti_1.2_Cr_1.2_Co_1.2_)O_19_	16.013	38.873	7.777	13.517	9.126	7.400	7.296

**Table 3 nanomaterials-09-00559-t003:** The chemical composition and average sample formulas.

#	Chemical Composition, wt. %							Sample Formula
	Ba	Fe	Ti	In	Ga	Cr	Co	
1	3.36	38.90	-	-	-	-	-	BaFe_12_O_19_
2	3.45	19.28	3.91	3.78	4.06	3.94	3.71	Ba(Fe_6_Ga_1.25_In_1.17_Ti_1.21_Cr_1.22_Co_1.15_)O_19_

**Table 4 nanomaterials-09-00559-t004:** Calculated unit cell parameters of barium hexaferrite.

No.	Synthesis Method	*a* [Å]	*c* [Å]	*V* [Å^3^]
[31]	BaFe_12_O_19_	5.893	23.194	697.5
1	BaFe_12_O_19_	5.8922 (1)	23.1953 (6)	697.40 (2)
2	Ba(Fe_6_Ga_1.25_In_1.17_Ti_1.21_Cr_1.22_Co_1.15_)O_19_	5.9253 (5)	23.5257 (22)	715.32 (9)

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
