# Peer review of "Extremely Polysubstituted Magnetic Material Based on Magnetoplumbite with a Hexagonal Structure: Synthesis, Structure, Properties, Prospects"

_nanomaterials, 2019, doi:10.3390/nano9040559_

Round 1
Reviewer 1 Report
Comment are given in the attached file.

Author Response
XRD
The Authors registered diffractogram for non-substituted and substituted sample. I
understood that they performed indexing finding cell parameters and ascribing Miller indexes
to diffraction lines. It is one of the most important part of this work proving that the obtained
compound is substituted magnetoplumbite. Therefore, special attention should be paid to give
as many details as it is possible in the Experimental part. There are missing information about
software and method used for cell parameters finding. Has been established also space group?
Answer: Corrected. The text was added:
The space group was established. The substituted sample and initial matrix has the same space group P6(3)/mmc.
It is not given if 10-80° corresponds to Q or 2Q.
Answer: 2Q was added.
Why the data collection started at 10° if
for c = 23.5257(22) Е and P6(3)/mmc the first reflection (002) should occur at ca. 7.6° 2Q?
If possible please repeat this experiment starting at 5° theta.
Answer: We are agree that first reflection at literature pattern was detected at 7.6° 2Q. But it has small intensity. We will take into account the Reviewer’s comment for future. But right now we have no opportunity to repeat the experiment starting at 5° theta (due to technical problem).
In „Results” they give several indices in three-number system and some in four-number
system. Please, select one system.
Answer: Done.
Why there are smaller reflection number at a diffractogram
2 than at a diffractogram 1? Why this number is substantially smaller than line number given
in the data available as reference patterns in databases (nevertheless, the main diffraction
lines are recorded). I would rather expect the opposite situation if doping would lower the
symmetry.
Answer: We think that it is due to the anisotropy. That is why the reflection intensity of substituted sample and initial matrix are different.
Samples.
The Authors mentioned powdered samples but they do not give any details (apart from p.77:
mortar) if their samples were grindered and grain size was somehow controlled. I mean
before sintering because for the final product such data are given. The SEM images show
that 0.1 mm grain size seems to be overestimated and they are much smaller.
Answer: We did not measure grain size of the mixture before sintering.
Electrodynamic properties
“The real and imaginary parts of magnetic permeability have now significant changes and
show a good qualitative and quantitative ratio.” I do not understand this sentence. Please tidy
this part and rewrite it. Are this sentence related to 2-12 or 10-18 GHz and data for sample 1
and sample 2 are compared?
Answer: done. Rewritten.
Language and minor problems:
Answer: done.
Reviewer 2 Report
The manuscript reports preparation and characterization of a high-entropy mixed-oxide phase with magnetoplumbite structure. Ba-hexaferrite-related ceramics, where half of Fe ions are substituted with the foreign ions Ti, Co, In, Ga, Cr (BaFe6(Ti,Co,In,Ga,Cr)6O19) were prepared with conventional solid-state sintering at high temperatures (1400 oC). The properties of this high-entropy phase were compared with properties of the unsubstituted Ba hexaferrite.
My first concern is compatibility of the manuscript with the scope of the Nanomaterials journal, which should be relevant to any field of study that involves nanomaterials. Besides, the manuscript represents simple comparison of the properties of the two samples, highly polysubstituted hexaferrite ceramics and unsubstituted ceramics, without any real discussion. The differences in (electrodynamic) properties of the two ceramics are not significant.
Minor comments:
How the samples were prepared for quantitative SEM EDXS analysis? Generally, the flat surface of the sample should be provided. The standard deviations of the analyses should be given. The standardless EDXS analysis certainly doesn’t enable accuracy justifying results to be given on 0.001 wt.%!
It is not clear, what represents Sample 1 / Sample 2 in Figs. 4-10.
Author Response
Minor comments:
How the samples were prepared for quantitative SEM EDXS analysis? Generally, the flat surface of the sample should be provided.
Answer: The special procedure was used for sample preparation for quantitative SEM/EDX analysis. To improve the measurement occurancy the sample powders were applied to the holder and flattened with a glass.
The standard deviations of the analyses should be given.
Answer: The standard deviations of the analyses was up to 5%.
The standardless EDXS analysis certainly doesn’t enable accuracy justifying results to be given on 0.001 wt.%!
Answer: Thank you for that comment. We are agree with the Reviewer. This accuracy (e.g. 3.912) was obtained by calculating the arithmetic mean value. The accuracy was revised.
It is not clear, what represents Sample 1 / Sample 2 in Figs. 4-10.
Answer: Sorry. It should be mentioned that Sample 1 is initial matrix BaFe12O19 and Sample 2 is substituted sample Ba(Fe6Ga1.25In1.17Ti1.21Cr1.22Co1.15)O19 (numbers 1 and 2 at Tables 2-4).
Round 2
Reviewer 1 Report
In the current version the manuscript can be published. All tasks were solved or explained adn these justifications complete my requirements.
Reviewer 2 Report
The authors successfully addressed minor comments. Still, my main concerns remain. The manuscript compares properties of two samples (which are similar). Without any in-depth scientific discussion I do not see how the results are relevant to the field.